# Validity of Actigraph for Measuring Energy Expenditure in Healthy Adults: A Systematic Review and Meta-Analysis

**DOI:** 10.3390/s23208545

**Published:** 2023-10-18

**Authors:** Wen-Jian Wu, Hai-Bin Yu, Wei-Hsun Tai, Rui Zhang, Wei-Ya Hao

**Affiliations:** 1School of Sports Science, Fujian Normal University, Fuzhou 350117, China; qsz20221035@student.fjnu.edu.cn; 2School of Physical Education, Quanzhou Normal University, Quanzhou 362000, China; zhangrui@jlu.edu.cn (R.Z.); haoweiya@ciss.cn (W.-Y.H.); 3Graduate School, Chengdu Sport University, Chengdu 610000, China; 4Key Laboratory of Bionic Engineering (Ministry of Education, China), Jilin University, Changchun 130022, China; 5China Institute of Sport Science, General Administration of Sport of China, Beijing 100061, China

**Keywords:** Actigraph, indirect calorimetry, energy expenditure

## Abstract

Purpose: The objective of this systematic review and meta-analysis was to assess the validity of the Actigraph triaxial accelerometer device in measuring physical activity energy expenditure (PAEE) in healthy adults, with indirect calorimetry (IC) serving as the validity criterion. Methods: A comprehensive search was conducted using the PubMed, Web of Science, and sportdiscuss databases, in addition to manual searches for supplementary sources. Search strategies were employed that involved conducting single keyword searches using the terms “gt3x” and “Actigraph gt3x”. The literature search encompassed the timeframe spanning from 1 January 2010 to 1 March 2023. The methodological quality of the studies included in the analysis was evaluated using both the Downs and Black checklist and the Consensus-Based Criteria for Selection of Measurement Instruments (COSMIN) checklist. The meta-analysis was conducted using the Review Manager 5.4 software. The standardized mean difference (SMD) was calculated and expressed as a 95% confidence interval (CI). The significance level was set at α = 0.05. A systematic assessment of the Actigraph’s performance was conducted through the descriptive analysis of computed effect sizes. Results: A total of 4738 articles were retrieved from the initial search. After eliminating duplicate articles and excluding those deemed irrelevant, a comprehensive analysis was conducted on a total of 20 studies, encompassing a combined sample size of 1247 participants. The scores on the Downs and Black checklist ranged from 10 to 14, with a mean score of 11.35. The scores on the COSMIN checklist varied from 50% to 100%, with an average score of 65.83%. The meta-analysis findings revealed a small effect size (SMD = 0.01, 95% CI = 0.50–0.52, *p* = 0.97), indicating no statistically significant difference (*p* > 0.05). Conclusions: The meta-analysis revealed a small effect size when comparing the Actigraph and IC, suggesting that the Actigraph can be utilized for assessing total PAEE. Descriptive analyses have indicated that the Actigraph device has limited validity in accurately measuring energy expenditure during specific physical activities, such as high-intensity and low-intensity activities. Therefore, caution should be exercised when utilizing this device for such purposes. Furthermore, there was a significant correlation between the activity counts measured by the Actigraph and the PAEE, indicating that activity counts can be utilized as a predictive variable for PAEE.

## 1. Introduction

Caspersen et al. [1] provided a definition for physical activity, stating that it encompasses “any bodily movement that leads to energy expenditure through the contraction of skeletal muscles”. A lack of physical activity has been identified as a contributing factor to the development of chronic diseases, including diabetes, coronary heart disease, and stroke [2]. Therefore, it is of the utmost importance to precisely assess levels of physical activity. Physical activity energy expenditure (PAEE) is the amount of energy expended during physical activity and is an important indicator of the physical activity level. The assessment of PAEE can provide valuable assistance to individuals or trainers in the development and modification of training programs, thereby enhancing their effectiveness. Additionally, PAEE measurement can contribute to weight management efforts [3]. For individuals diagnosed with metabolic disorders, such as diabetes, having a comprehensive understanding of their PAEE can greatly assist healthcare professionals in effectively managing their condition [4]. Furthermore, comprehending the concept of PAEE in the human body can offer researchers significant insights into health and nutrition-related matters [5]. As the field of science and technology advances, an increasing number of techniques are being developed to measure PAEE. Measurements of PAEE encompass various methods, such as indirect calorimetry, doubly labeled water, heart rate monitoring, questionnaires, and accelerometers. Indirect calorimetry (IC) and doubly labeled water (DLW) are two widely recognized and reliable methods of assessing PAEE [6,7,8]. These methods are considered the gold standard in the field [9]. While both IC and DLW methods have the ability to accurately measure PAEE, it is important to note that these methods often necessitate the use of specialized equipment and laboratory set-ups. This includes the requirement for gas analyzers, metabolic instrumentation, and experiments involving labeled isotopes. The cost of the equipment and the need for specialized skills to operate it render these methods less viable for conducting extensive measurements of energy expenditure or for long-term monitoring in real-life scenarios. When selecting a method for assessing PAEE, several factors need to be taken into account. These factors include cost-effectiveness, portability, and measurement accuracy. Accelerometers are widely utilized as measurement instruments for the estimation of PAEE due to their cost-effectiveness, portability, and high level of accuracy [10,11]. The Actigraph has a substantial and extensive background in the realm of physical activity measurement. Consequently, its equipment has undergone numerous iterations and enhancements to cater to diverse research requirements. Among the available options, the Actigraph triaxial accelerometer has emerged as the most extensively utilized device in numerous studies focused on monitoring human PAEE [12]. The Actigraph triaxial accelerometer is achieved by employing microelectromechanical system (MEMS) sensors that are oriented along three orthogonal axes, namely vertical, coronal, and sagittal. The sensor is capable of detecting changes in motion and converting them into digital signals, which can then be analyzed to estimate energy expenditure. The device generates activity counts for each axis and a composite vector magnitude (VM) that encompasses all three axes [13,14]. Software algorithms can be employed to compute energy expenditure or metabolic equivalents by utilizing the raw counts obtained during physical activities. Actigraph sensors have a broad range of applications across various disciplines, such as health sciences, medicine, psychology, exercise science, and epidemiology. Actigraph triaxial accelerometers play a crucial role in investigating the intricate connections between physical activity and health. By providing objective data, they contribute to health assessment and the formulation of interventions.

Studies have conducted systematic reviews to assess the accuracy of the Actigraph in calculating step counts [15]. The findings from these evaluations indicate that the Actigraph demonstrates exceptional accuracy in step counting. Nevertheless, a comprehensive analysis or systematic review of existing studies examining the accuracy of the Actigraph in quantifying PAEE has not been found. Therefore, it is imperative to conduct additional systematic and comprehensive studies in order to thoroughly evaluate the precision and dependability of the Actigraph’s performance in measuring PAEE. The aim of this study was to assess the accuracy of the Actigraph as a tool for quantifying PAEE in a population of healthy individuals, using a meta-analysis and systematic review approach. A meta-analysis is a statistical technique that combines the findings of multiple studies to generate a comprehensive effect estimate. This approach enhances the study’s ability to draw more comprehensive conclusions, thereby strengthening its overall validity and reliability. This study aims to utilize IC as a validity criterion in order to assess the accuracy of the Actigraph’s measurement of PAEE in a population of individuals who are in good health. Given that IC is widely acknowledged as a dependable indicator of PAEE, employing IC as a criterion for validity yields more consistent and trustworthy outcomes compared to alternative approaches that may lack sufficient accuracy. This characteristic represents a notable advantage of the present study. In conclusion, this study makes a valuable contribution to the assessment of the Actigraph’s reliability as a tool for measuring physical activity. It also provides policymakers, healthcare professionals, and researchers with essential information to enhance their understanding and evaluation of the effects of physical activity on health. This can contribute to enhancing the knowledge base for health interventions, disease management, and public health policy.

## 2. Methods

This study adhered to the guidelines outlined in the Preferred Reporting Items for Systematic Reviews and Meta-Analyses (PRISMA) (Appendix A) [16]. However, it is important to note that the program for this systematic review and meta-analysis was not registered.

### 2.1. Search Strategy

The databases PubMed, Web of Science, and sportdiscuss were all systematically searched, with additional manual searches conducted to supplement the results. The search was conducted within the timeframe of 1 January 2010 to 1 March 2023. In the initial stages, we conducted a combinatorial search using Boolean logic methodology. However, it became evident that the application of Boolean logic resulted in a limited range of literature to be examined. In the final analysis, a collective decision was reached to conduct the search using individual keywords. The search strategy and outcomes are shown in Table 1.

### 2.2. Eligibility Criteria

The studies included in this analysis were required to meet specific criteria. (1) They had to involve the simultaneous use of both IC and Actigraph triaxial accelerometers in the experiments. (2) The studies needed to analyze the differences, correlations, and agreements between the Actigraph device and IC. (3) The participants had to be in a healthy state, capable of moving without assistance, and without any contraindications to physical activity. (4) The studies had to be published in English. By exclusively selecting healthy populations for the study, researchers can enhance their control over potential confounding factors. Non-healthy populations often exhibit a range of chronic illnesses, medication usage, and other health complications that can impede the accurate measurement of energy expenditure. Excluding these factors can contribute to enhancing the representativeness and reflectiveness of the Actigraph triaxial accelerometer’s performance in the study. Therefore, studies that included non-healthy populations were also excluded. Two researchers conducted the literature screening process independently, and any discrepancies that arose were resolved through collaborative discussion.

### 2.3. Methodological Quality

Methodological quality assessment, specifically evaluating the risk of bias, is a crucial preliminary step in conducting a meta-analysis study. In the present study, a modified edition of the Downs and Black checklist was employed to evaluate the methodological quality. A modified iteration of the Downs and Black checklist has been deemed appropriate for the evaluation of non-randomized controlled trials [17]. This checklist has been effectively utilized in previous studies involving systematic reviews [18,19]. The evaluation tool consists of seventeen items, each with a maximum score of 17. Eight items were used to evaluate the quality of the reporting guidelines, while three items were used to assess external validity and five items were used to assess internal validity. Studies with scores ranging from 13 to 17 are deemed to be of high quality, whereas studies with scores below 13 but equal to or greater than 9 are categorized as moderate quality. Studies with scores below 9 are considered to be of low quality [20].

To augment the accuracy of the assessment of bias, an additional evaluation was performed on studies that examined measurement instruments reported by proxies. This assessment employed the Consensus-Based Criteria for Selection of Measurement Instruments for Health (COSMIN) risk of bias checklist to identify any potential additional biases. The COSMIN framework serves as a methodological tool and guide for assessing health measurement instruments. The primary aim of the checklist is to assist researchers and clinical practitioners in the selection, evaluation, and utilization of appropriate health measurement instruments, thereby guaranteeing the quality and dependability of their findings [21,22]. The COSMIN checklist has been developed to evaluate multiple characteristics of measurement instruments, such as reliability, validity, sensitivity, specificity, and other pertinent factors. However, it is imperative to acknowledge that the seventh item of the COSMIN checklist is not applicable in this particular case. This is due to the fact that Actigraph triaxial accelerometer measurements yield continuous scores rather than binary ratings. Among the six items, a point is granted upon meeting the conditions, whereas no points are given if the conditions are not fulfilled. The maximum achievable score for this item is 6. The COSMIN score was ultimately converted into a percentage by calculating the ratio of the obtained score to the total score. According to the study conducted by [23], the classification of quality scores was as follows: a score of 76–100% was categorized as high quality, 51–75% as medium quality, 26–50% as low quality, and 0–25% as very low quality. After a thorough and unbiased evaluation carried out by two independent reviewers, a subsequent meeting was convened to reach a consensus on the assessment findings.

### 2.4. Data Extraction

Data extraction was carried out independently by two researchers using the Microsoft Excel software. The information that was extracted encompassed various variables, such as the authors, sample size, gender, age, body mass index (BMI), devices used, wearing position of Actigraph devices, sampling frequency, measured outcomes, and validity indices. The extracted validity indices included measures of accuracy, simple correlation coefficients (Pearson and Spearman), intraclass correlation coefficients, and confidence intervals.

### 2.5. Data Analysis

Energy expenditure data were obtained from existing literature sources. In certain studies, energy expenditure for various physical activities was assessed individually. Subsequently, the data were extracted and the energy expenditure for each physical activity was aggregated using Excel in order to calculate the overall energy expenditure for the physical activity. Given the inherent challenges associated with comparing various forms of physical activity, a meta-analysis was conducted using the Review Manager 5.4.1 software to analyze the total energy expenditure measured by the Actigraph and indirect calorimetry, to conduct a comparative analysis of the combined effect sizes of the Actigraph and IC. Given that the study’s outcome indices were continuous variables, the effect size was measured using the standardized mean difference (SMD). Additionally, 95% confidence intervals were computed with a significance level of α = 0.05. The heterogeneity test can assist researchers in selecting a suitable statistical model to conduct a meta-analysis. When the value of I^2^ is less than or equal to 50%, it is deemed that there is no substantial statistical heterogeneity, and a fixed-effects model is employed. Conversely, when the value of I^2^ exceeds 50%, it indicates the presence of statistical heterogeneity among the studies, and a random-effects model is recommended [24]. Sensitivity analysis holds significant importance in the field of meta-analysis. Sensitivity analysis facilitates a comprehensive understanding of the stability of the aggregated outcomes, enabling the determination of whether specific choices or model assumptions have an impact on the results. In the present study, a sensitivity analysis was conducted by substituting the random-effects model with a fixed-effects model. If the calculated results exhibited consistency across both models, it indicated a higher level of reliability in the study findings.

Effect sizes serve as metrics utilized to quantify the magnitude of observed effects within a study. For the purpose of conducting a descriptive analysis, the SPSS Statistics 26 software was utilized to determine the effect size value for each physical activity. The effect sizes were utilized to assess whether there existed a statistically significant distinction between the measured effects of the two measures. The correlation coefficients and criteria for evaluating the effect size are presented in Table 2. If the correlation coefficient was determined to be of high quality or exceptional, and the effect size was found to be minimal, the device was deemed valid.

## 3. Results

A total of 4723 articles were obtained from the three databases, supplemented by an additional 15 articles discovered through manual searches. Duplicate removal was conducted using Zotero 5.0.94, an open-source reference management software program developed by the Roy Rosenzweig Center for History and New Media in Virginia, United States. This process yielded a total of 1184 distinct articles. In the final analysis, a total of 20 studies met the eligibility criteria, as depicted in the screening flowchart (refer to Figure 1).

### 3.1. General Characteristics of the Studies

The current study conducted a comprehensive analysis of 20 scholarly articles, encompassing a participant cohort of 1247 individuals. Among the studies included in this analysis, a total of 16 studies [28,29,30,31,32,33,34,35,36,37,38,39,40,41,42,43] diligently provided information on the average age of the participants. The mean age, calculated meticulously across the amalgamated studies, was determined to be 27.68 ± 17.99 years. Furthermore, thirteen studies [28,29,30,31,32,33,34,35,37,38,40,42,43] provided valuable insights into the body mass index (BMI) of 432 participants. The mean body mass index (BMI) was accurately calculated and found to be 23.05 ± 0.22 kg/m^2^ (refer to Table 3).

### 3.2. Methodological Quality

The studies included in the analysis achieved scores on the Downs and Black checklist that ranged from 10 to 14, with an average score of 11.35. Two articles were assessed as high quality [32,45], whereas the remaining studies were categorized as medium quality [28,29,30,31,33,34,35,36,37,38,39,40,41,42,43,44,46,47] (Appendix A).

The scores for the COSMIN checklist were reported as percentages, ranging from 50% to 100%, with an average score of 65.83%. Among the studies that were included, four [31,32,41,45] were assessed as high quality, while ten [28,29,30,36,38,39,42,43,44,46] were categorized as medium quality. Additionally, six studies [33,34,35,37,40,47] were deemed to be of low quality (Appendix A).

### 3.3. Data Analysis

The 20 articles included in this study were categorized into three groups. The first category examined the relationship between Actigraph activity counts [28,29,30,31] and indirect calorimetry, with articles [28,29,30,31] focusing on this topic. The second category involved a comparison of energy expenditure measurements between the Actigraph and indirect calorimetry [32,33,34,35,36,37,38,41,42,43] and a comparison of different energy expenditure calculation equations [39,40,44,45,46,47].

#### 3.3.1. Correlation between Activity Counts and Energy Expenditure

A comprehensive analysis was conducted on four articles to examine the relationship between the vertical axis (VA), vector magnitude (VM) activity counts, and indirect calorimetry. The findings of these studies are summarized in Table 3 [28,29,30,31]. Neil-Sztramko et al. [29] and Kemp et al. [31] did not provide information regarding the specific counting method employed for the correlation analysis. Kelly et al. [28] conducted a correlation analysis using VM counts, while Hänggi et al. [30] examined the correlation between both VA and VM. Three articles investigated the association between total energy expenditure and reported Pearson correlation coefficients ranging from 0.69 to 0.89 [28,29,30]. One scholarly article conducted a separate analysis to examine the correlations between various activities. The study reported Spearman correlation coefficients ranging from 0.32 to 0.59 [31] (refer to Table 4).

#### 3.3.2. Validity of Actigraph Devices

##### Meta-Analysis Results

To minimize heterogeneity across studies, a meta-analysis was performed on a total of eight studies [33,34,35,36,38,41,42,43] in which participants wore the Actigraph device on their hip. The I^2^ statistic, with a value of 85%, suggested a substantial level of heterogeneity (*p* < 0.001). Consequently, a random-effects model was utilized for the analysis. The SMD is a result calculated during the process of meta-analysis, which represents the effect size between the Actigraph and IC. With the use of the SMD, a meta-analysis has the ability to consolidate various effect sizes into a single unified metric, facilitating combined analysis and comparison. The findings indicated that the combined effect size was SMD = 0.01, with a 95% confidence interval ranging from −0.50 to 0.52 and a *p*-value of 0.97, which is greater than the significance level of 0.05. This finding indicates that there was no statistically significant disparity observed between Actigraph and IC in their ability to measure the total PAEE (Figure 2).

##### Sensitivity Analysis

The meta-analysis conducted on measurements of total energy expenditure revealed a significant level of heterogeneity (I^2^ = 85%, *p* < 0.00001). To evaluate the sensitivity of the analysis [33,34,35,36,38,41,42,43], the random-effects model was replaced with a fixed-effects model. The sensitivity analysis revealed that the findings were consistent with the fixed-effects model (SMD = −0.01, 95% confidence interval [CI]: −0.21 to 0.18, *p* = 0.88 > 0.05). This statement suggests that there were no significant alterations in the fundamental findings between the two models, thereby establishing the stability and reliability of the meta-analysis results (Figure 3).

##### Descriptive Analysis

Among the studies included in the analysis, three papers [32,34,38] utilized the Pearson correlation coefficient to assess the association between the Actigraph and IC. Additionally, three papers [34,42,43] used the intra-class correlation coefficient for this purpose. Furthermore, eight papers [33,34,35,36,37,38,41,43] compared the means using different measures, including the mean deviation, percentage error, mean percentage error, root mean square error, and mean percentage absolute error. Two papers [32,38] exclusively presented the effect size for overall activity, whereas eight papers [33,34,35,36,37,41,42,43] further subdivided the effect size for various activities (refer to Table 5).

#### 3.3.3. Accuracy of Different Prediction Equations

The calculation formula utilized is presented in Table 6. A comprehensive analysis was conducted, encompassing a total of six articles, which aimed to compare the accuracy of various equations in predicting PAEE for identical activities or the same equation for different activities [39,40,44,45,46,47]. Among them, four articles [39,44,45,46] provided access to raw data that allowed for the calculation of effect sizes. These effect sizes ranged from 0.16 to 6.34. The two remaining articles [40,47] were unable to obtain raw values but instead reported the data bias and mean absolute percentage error, as shown in Table 7.

## 4. Discussion

### 4.1. The Correlation between Counts and Energy Expenditure

The previous studies have verified the reliability of activity counting for three-axis accelerometers, namely the Actigraph GT3X, GT3X+, and wGT3X-bt, using a vibration table [56]. It has been determined that triaxial accelerometers exhibit higher levels of accuracy compared to single-axis accelerometers. Examining the relationship between activity counts and IC is a crucial requirement in evaluating the reliability of the Actigraph. This correlation is essential in determining whether the Actigraph can be effectively utilized as a means of assessing PAEE.

Table 4 shows that there is a significant moderate to high correlation between Actigraph triaxial accelerometer-derived activity counts and IC when all types of physical activity are measured together [28,29,30]. Activity counts derived from Actigraph accelerometers can serve as a valid variable for predicting PAEE. However, we also found no correlation between activity counts and energy expenditure when using the Actigraph for a particular physical activity. For example, activity counts did not significantly correlate with energy expenditure when wearing the Actigraph triaxial accelerometer during dynamic cycling, resting, standing, walking at a moderate speed, and fast running activities [31]. We hypothesize that this lack of significant correlation may be due to the fact that stationary cycling, resting, and standing activities do not produce significant spatial acceleration, resulting in small accelerometer readings that cannot be correlated with energy expenditure.

In summary, the aforementioned findings indicate that the Actigraph three-axis accelerometer can serve as a reliable measure for predicting energy expenditure during physical activity in a general context. However, it is important to note that its accuracy may be constrained in certain specific activity scenarios. Therefore, it is imperative to consider activity types and conditions when utilizing the Actigraph for research or evaluation purposes. This is crucial in order to determine its applicability and it is necessary to exercise caution when interpreting the results.

### 4.2. Validity According to the Actigraph for Energy Expenditure

The findings of the meta-analysis suggest that Actigraph triaxial accelerometers are a valid alternative to indirect calorimetry for measuring total PAEE during various activities. An intriguing observation can be made based on the data presented in Table 5: the Actigraph device may not consistently provide accurate measurements of energy expenditure during physical activity [36,41,43]. In contrast, the Actigraph proves to be valid when single activities are combined [36,41,43]. We hypothesize that the observed phenomenon could be attributed to the compensatory effect of potential overestimates and underestimates of energy expenditure in various activities. This compensation mechanism may help to mitigate the bias in the accelerometer’s assessment of total activity energy expenditure. From a theoretical standpoint, the Actigraph demonstrates the capacity to accurately assess the energy expenditure associated with physical activity across a range of activities over a specified duration.

In the study by Calabró et al. [32], we found that although the energy expenditure calculated using the Actigraph correlated significantly with IC, it significantly underestimated energy expenditure for activities such as walking, standing, and light housework. In the study conducted by Schneller et al. [33], it was observed that the Actigraph triaxial accelerometer exhibited a tendency to underestimate energy expenditure during low-intensity activities. Other studies [34,35] also indicated that the validity and errors in energy expenditure estimation by the device varied depending on the activity type. As previously stated, it is essential to have a strong correlation between acceleration counts and PAEE in order to ensure precise measurements. When examining the situation from this particular standpoint, it is hypothesized that insufficient spatial acceleration during low-intensity exercise may lead to a reduced correlation between acceleration counts and PAEE. In Table 5 [36,40], it can be observed that the effect size between IC and the Actigraph during running at speeds of 6.5–9.5 km/h is less than 0.2, indicating that the Actigraph is valid and has higher measurement accuracy in the situation.

Small fluctuations in acceleration can lead to inaccuracies in the Actigraph’s measurements during low-intensity sports. The performance of the Actigraph can also be called into question when used in high-intensity sports. Gastin et al. [37] observed a significant effect size between the Actigraph and IC during high-intensity interval training, suggesting that the Actigraph exhibits larger deviations. This particular observation was also documented by Morris et al. [38], who discovered that while the Actigraph yielded measurements that were relatively close to those obtained through IC in comparison to other monitors, the device did not demonstrate sufficient accuracy for the purpose of quantifying energy expenditure during high-intensity training. We hypothesize that the bias observed during high-intensity exercise may be attributed to rapid fluctuations in acceleration, which could potentially surpass the maximum measurement range of the equipment. This is in contrast to activities with minimal changes in spatial acceleration [33,36].

In conclusion, the Actigraph demonstrated validity in measuring PAEE for overall activity. However, the study showed some degree of bias in assessing PAEE for individual activities. The relationship between acceleration counts and energy expenditure showed variability across different types of physical activity. Therefore, careful consideration must be given to the type and intensity of activity when using the Actigraph for PAEE estimation. The effects of the activity type and intensity on the accuracy of Actigraph measurements should be explored in future studies to enhance our understanding of the specific activity types and conditions in which the Actigraph may perform poorly and to identify ways to enhance its performance. These endeavors will contribute to the advancement of our comprehension regarding the viability and constraints of the Actigraph as a measurement tool for PAEE. Furthermore, they will aid in the optimization of its utilization across various research and application domains.

### 4.3. Accuracy of Different Prediction Equations

If there is a significant correlation between the Actigraph’s activity counts and energy expenditure, the accuracy of its measurements is contingent upon the prediction equation used. The prediction equations commonly utilized are presented in Table 6. Different populations exhibit distinct characteristics during physical activities, necessitating the development of prediction equations tailored to each population. The study conducted by Crouter et al. [39] aimed to assess PAEE in a sample of children. The findings indicate that the effect size of Puyau’s [52] energy expenditure equation is less than 0.2, suggesting that Puyau’s equation [52] is a valid measure for estimating PAEE in children. However, Zhu et al. [44] conducted a study that revealed that there is no singular equation that can accurately predict the energy expenditure associated with physical activity in children. In 2015, Crouter et al. [45] developed two novel equations, REG-VA and REG-VM, which aimed to assess physical activity levels in children. These equations were formulated based on activity counts obtained from wrist-worn devices. It has been confirmed that these equations are valid [45]. For the elderly population, it is necessary to utilize specific formulas in order to accurately estimate PAEE. This is due to the fact that their activity patterns and metabolic rates may vary from those observed in younger adults. However, an examination of Table 7 reveals that the effect size values for various equations are all greater than 0.2 [40,46,47], which indicates that none of the equations available are able to accurately measure the energy expenditure of the elderly population.

Predictive equations have a significant role in the measurement of PAEE, and various populations have specific equations that are applicable to them. Therefore, it is imperative to meticulously consider the characteristics of the study population when choosing and implementing prediction equations in order to obtain more precise estimations of energy expenditure. As previously mentioned, it is important to note that various forms of physical activity exhibit distinct acceleration characteristics. Therefore, it is recommended that in future studies, calculation equations should be devised and chosen accordingly, taking into consideration the specific type of physical activity being analyzed.

### 4.4. Validity According to the Positioning of Actigraph Devices

The importance of activity counts for the accurate measurement of PAEE is already known, with different wearing positions resulting in different activity counts for the same activity. Several studies have underscored the significance of anatomical positioning in the precise assessment of energy expenditure during physical activity [47,49,57,58,59].

Some studies [41,42,46,60,61] have compared different wearing positions (Table 5). In the study by Mcminn et al. [41], it was found that although the effect size was always greater than 0.2, the device had a smaller effect size when worn on the hip during jogging and a smaller effect size when worn on the wrist during fast running. Moreover, in the study by Kossi et al. [42], they also found different effect sizes for different types of activities and different wearing positions. In addition to the anatomical position, the choice of the side on which the device is worn can also have an impact on activity counts. However, the study demonstrated a notable disparity in activity counts when the device was worn on the left or right wrist, while no significant difference in activity counts was observed when the device was worn on the left or right hip [60]. The choice of wearing position should be based on the characteristics of the physical activity. Wearing the device on the upper limb to measure lower limb movement or on the lower limb to measure upper limb movement [61] can result in significant inaccuracies. It has been suggested that the selection of the wearing position should be determined by the specific activity [46].

From the descriptive analysis presented in Table 5, it is evident that the impact of wearing the device on the hip for measuring total energy expenditure per activity is small [62], while the findings from the meta-analysis suggest that wearing the Actigraph on the hip for measuring total energy expenditure is indeed effective. Therefore, it is advisable to wear the device on the hip while measuring total energy expenditure. When measuring a single activity, such as cycling or stair climbing, it is important to select the appropriate wearing position based on the specific characteristics of the activity.

### 4.5. Practical Implications and Future Trends

This study has conducted an assessment of the Actigraph’s validity in measuring PAEE within a healthy population. The results of the study are encouraging and show that there is no significant difference between the Actigraph and IC, which means that we can use the Actigraph as a reliable method to predict PAEE. This finding is significant because it provides us with a powerful tool to gain insights into and optimize an individual’s physical activity level and overall health. By utilizing the Actigraph to estimate PAEE, we are able to more accurately assess the amount of energy required by an individual to engage in physical activity. This has important implications for both researchers and healthcare professionals. It means that we can more accurately assess the potential health benefits of various activities and therefore better customize health and exercise programs for individuals to suit their needs. While the results generally support its validity, certain physical activities yielded less impressive measurements. The findings of the study serve as a reminder to take into account the influence of the specific type of activity when assessing PAEE. It is of the utmost importance to utilize the Actigraph device in accordance with the specific attributes of the activity.

PAEE is recognized to exhibit variability across different activity types. Therefore, it is recommended that future research focuses on developing formulas that account for the activity type rather than solely relying on population-based approaches. In the present investigation, it was also observed that there is a limited number of studies examining the energy expenditure associated with slope motion using the Actigraph [43]. Therefore, it is necessary to conduct further research to assess the accuracy and reliability of the Actigraph in measuring energy expenditure during slope motion.

## 5. Strengths and Limitations

Regarding the limitations of this study, it is noteworthy that there are other triaxial accelerometers available in the market, manufactured by different companies, which are utilized for comparable activity measurements. More comprehensive validation and comparative studies are required to assess the validity and reliability of these competing products. Future research endeavors may contemplate broadening the scope of investigation to encompass a diverse range of triaxial accelerometers, including multiple brands and models. This would enable a comprehensive evaluation of their performance across different environmental conditions. Additionally, in the context of this study, the primary objective was to evaluate the validity of the Actigraph as a measure of energy consumption by comparing it to indirect calorimetry, which is widely regarded as the “gold standard” method. However, it is worth considering the comparison of the Actigraph with another frequently employed method of measuring energy consumption, namely the doubly labeled water method. The doubly labeled water method is a precise and reliable technique for quantifying energy expenditure. This method involves monitoring the metabolism and excretion of metabolic markers within the body to estimate energy expenditure accurately. Comparing the Actigraph to the doubly labeled water method offers supplementary validation of its accuracy and contributes additional data and references to the field of energy expenditure measurement.

Regarding the strengths of this study, combining a systematic review with a meta-analysis offers a more comprehensive and reliable approach to validate the Actigraph’s effectiveness in measuring PAEE. This approach has facilitated a more comprehensive understanding of the performance attributes of the Actigraph and its suitability in diverse scenarios. Indirect calorimetry is widely acknowledged as the most accurate and reliable technique for measuring energy expenditure, as it allows for the direct assessment of the physiological processes involved in energy metabolism within the human body. Comparing the Actigraph to an indirect calorimeter and evaluating its performance against the most reliable measurement methodology enhances the precision and dependability of measurements employed in the assessment of the Actigraph.

## 6. Conclusions

The main objective of this study was to test the validity of Actigraph triaxial accelerometers in measuring PAEE in healthy individuals, using indirect calorimetry as a reference standard. In the research, the meta-analysis, which was based on the total energy expenditure of each physical activity, showed that the effect size between the two groups was very small with Actigraph triaxial accelerometers compared to indirect calorimetry. It can be concluded that using an Actigraph to measure the total energy expenditure of physical activity is valid. The findings from the descriptive analyses suggest that the Actigraph triaxial accelerometer may not always provide accurate measurements of energy expenditure for single physical activities. The Actigraph is not proficient in accurately assessing energy expenditure for activities of both low and high intensity. Hence, it is advisable to exercise caution when utilizing the Actigraph as a means to quantify energy expenditure during a specific physical activity. Furthermore, there was a significant correlation between the activity counts measured by the Actigraph and the PAEE, indicating that activity counts can be utilized as a predictive variable for PAEE.

## Figures and Tables

**Figure 1 sensors-23-08545-f001:**
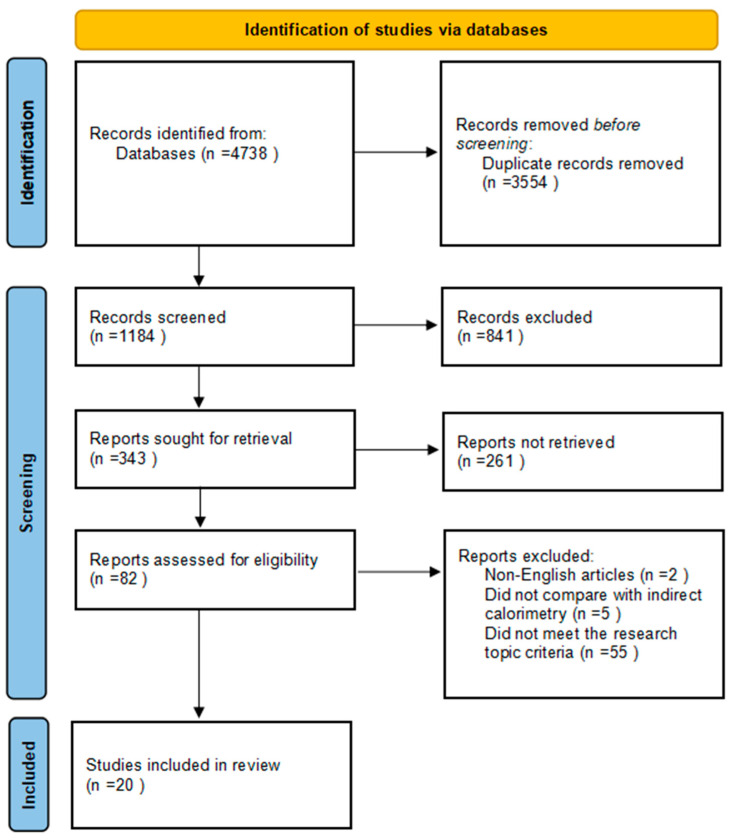
PRISMA flowchart of included studies.

**Figure 2 sensors-23-08545-f002:**
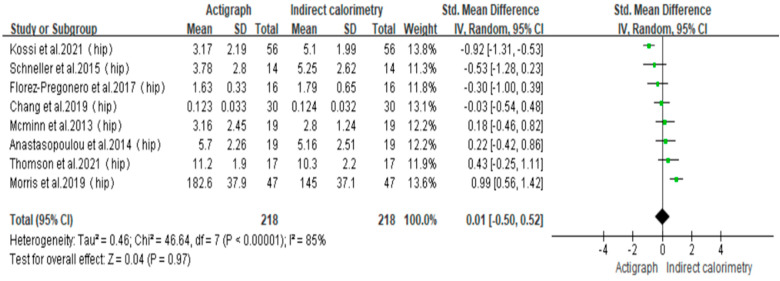
Forest plot [33,34,35,36,38,41,42,43].

**Figure 3 sensors-23-08545-f003:**
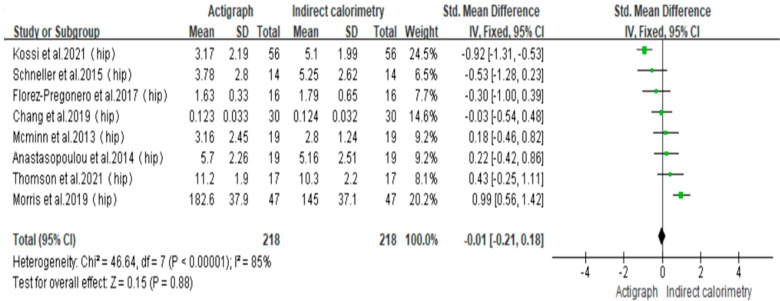
Sensitivity analysis [33,34,35,36,38,41,42,43].

**Table 1 sensors-23-08545-t001:** Search strategy and outcomes.

Source of Search	Date of Search	Searches Conducted	Search Outcome
PubMed	1 January 2010 to 3 January 2023	ALL-Fields (“gt3x”)	1048
		ALL-Fields (“Actigraph gt3x”)	965
Web of Science—Core	1 January 2010 to 3 January 2023	ALL-Fields (“gt3x”)	1045
		ALL-Fields (“Actigraph gt3x”)	951
sportdiscuss	1 January 2010 to 3 January 2023	ALL-Fields (“gt3x”)	390
		ALL-Fields (“Actigraph gt3x”)	324
Manual Search	1 January 2010 to 3 January 2023	No application	15

**Table 2 sensors-23-08545-t002:** Criterion of validity.

Interpretation of Pearson and Spearman Correlation Coefficients Using Cohen’s Criteria [25]	Intra-Group Correlation Coefficients Interpreted Using Ciccetti’s Criteria [26]	Explaining Effect Sizes Using Hopkins’ Criteria [27]
<0.3: Very low 0.3–0.49: Moderate 0.5–0.69: Good 0.7–1: Excellent	<0.4: Very low 0.4–0.59: Low 0.6–0.74: Good 0.75–1: Excellent	<0.2: Very low 0.2–0.59: Low 0.6–1.19: Moderate 1.2–1.99: Important 2–4: Very Important >4: Extremely important

**Table 3 sensors-23-08545-t003:** General characteristics of the studies.

Authors and Year	Number of Participants	Participants’ Age (Mean ± Standard Deviation), Years	Body Mass Index (Mean ± Standard Deviation), kg/m^2^	Actigraph Device	Positioning of Actigraph	Sampling Frequency	Indirect Calorimetry Device
Florez-Pregonero et al., 2017 [35]	16	25.38 ± 8.58	24.6 ± 4.6	NR	Hip	30 Hz	Oxycon mobile
Gastin et al., 2018 [37]	26	21.3 ± 2.4	23.2 ± 2.0	GT3X+	Hip	100 Hz	Metamax 3B
Calabró et al., 2014 [32]	40	27.4 ± 6.7	22.9 ± 3.1	GT3X	NR	NR	Oxycon mobile
Morris et al., 2019 [38]	47	28.12 ± 11.22	23.68 ± 3.97	GT3X+	Hip	NR	Cosmed k4b2
Kelly et al., 2013 [28]	42	21.57 ± 2.73	25.26 ± 3.25	GT3X	Hip	30 Hz	NR
Crouter et al., 2013 [39]	72	12.7 ± 0.8	NR	GT3X+	Hip	30 Hz	Cosmed k4b2
Hänggi et al., 2013 [30]	49	10.8 ± 1.9	18.2 ± 2.7	GT3X	Hip	NR	Cosmed k4b2
Schneller et al., 2015 [33]	14	27.7 ± 3.3	22.9 ± 1.4	GT3X+	Hip	60 Hz	Cosmed k4b2
Anastasopoulou et al., 2014 [34]	19	30.68 ± 8.58	24.34 ± 3.27	GT3X	Hip	30 Hz	Metamax 3B
Kossi et al., 2021 [42]	29	24 ± 4	25 ± 3	GT3X+	Hip, ankle and wrist	30 Hz	Metamax 3B
Zhu et al., 2013 [44]	367	NR	NR	GT3X	Hip	30 Hz	Cosmed k4b2
Aguilar-Farias et al., 2019 [40]	40	77.4 ± 8.13	22.0 ± 5.67	GT3X+	Hip	30 Hz	Metamax 3B
Chang et al., 2019 [43]	30	24.53 ± 1.55	23.86 ± 2.67	GT3X	Hip	30 Hz	Vmax encore 29 system
Crouter et al., 2015 [45]	181	NR	NR	GT3X	Wrist	30 Hz	Cosmed k4b2
Kemp et al., 2020 [31]	50	29.5 ± 18.0	24.1 ± 5.5	GT3X	Hip	NR	Cosmed cpet
Mcminn et al., 2013 [41]	19	30 ± 9	NR	GT3X+	Hip and wrist	30 Hz	Ultima cpx
Neil-Sztramko et al., 2017 [29]	30	40.0 ± 14.9	22.4 ± 3.1	GT3X+	Wrist	NR	Cosmed k4b2
Thomson et al., 2021 [36]	20	28 ± 5	NR	GT3X+	Hip	30 Hz	Ultima cpx
Kim et al., 2016 [46]	59	NR	NR	GT3X	Hip	30 Hz	Oxycon mobile
Santos-Lozano et al., 2013 [47]	97	NR	NR	GT3X	Hip	30 Hz	Oxycon pro

NR = not reported.

**Table 4 sensors-23-08545-t004:** Correlation between activity counts and indirect calorimetry.

Authors	Activity Counts (Mean ± Standard Deviation)	Indirect Calorimetry (Mean ± Standard Deviation)	Validity Indices	Outcomes
Kelly et al., 2013 [28]	VM 4.8 km/h:5688.83 ± 1072.3 (cpm) 6.4 km/h:8470.14 ± 1402.9 (cpm) 9.7 km/h:12774.0 ± 2413.8 (cpm)	4.8 km/h:0.93 ± 0.2 (L/min) 6.4 km/h:1.32 ± 0.3 (L/min) 9.7 km/h:2.54 ± 0.5 (L/min)	Pearson correlation coefficient	VM r = 0.81 * Type of activity: 4.8 km/h slow walk, 6.4 km/h fast walk, 9.7 km/h run
Hänggi et al., 2013 [30]	VA Lying: 0.08 ± 0.25 (cps) Sitting: 0.04 ± 0.19 (cps) Stand: 0.83 ± 1.47 (cps) Video game: 5.33 ± 9.76 (cps) Slow walking: 28.07 ± 8.96 (cps) Brisk walking: 62.09 ± 13.13 (cps) Jogging: 122.09 ± 30.13 (cps) Moderate run: 122.50 ± 33.98 (cps) VM Lying: 0.24 ± 0.54 (cps) Sitting: 0.26 ± 0.83 (cps) Standing: 3.79 ± 4.2 (cps) Video game: 26.34 ± 23.16 (cps) Slow walking: 44.95 ± 9.63(cps) Brisk walking: 81.32 ± 13.12(cps) Jogging: 136.34 ± 28.51(cps) Moderate running: 140.73 ± 33.83(cps)	Lying: 5.43 + 1.46 (mL∙kg^−1^/min) Sitting: 5.97 + 1.46 (mL∙kg^−1^/min) Standing: NR Video game: 12.20 + 5.15 (mL∙kg^−1^/min) Slow walking: 14.8 + 4.65 (mL∙kg^−1^/min) Brisk walking: 20.57 + 5.04 (mL∙kg^−1^/min) Jogging: 29.78 + 5.97 (mL∙kg^−1^/min) Moderate running: 34.35 + 6.59 (mL∙kg^−1^/min)	Pearson correlation coefficient	VA r = 0.88 * VM r = 0.89 * Type of activity: lying, sitting, standing, boxing, walking, and running
Neil-Sztramko et al., 2017 [29]	NR Sitting: 504.2 ± 703.7 (cpm) Carrying: 6828.4 ± 1548.7 (cpm) Climbing stairs: 6735.5 ± 2720.1 (cpm) 2 mph Jogging: 2816.6 ± 1013.3 (cpm) Self-selected speed jogging: 4718.4 ± 1955.5 (cpm) 3.0–3.5 mph medium-speed run: 5432.2 ± 1853.3 (cpm) Optional medium-speed run: 6642.6 ± 1975.0 (cpm) Fast run: 11,235.4 ± 6856.5 (cpm) Optional fast-speed run: 11585.5 ± 6297.3 (cpm)	Sitting: 1.4 ± 0.3 (MET) Carrying: 5.8 ± 1.3 (MET) Climbing stairs: 7.9 ± 1.5 (MET) 2 mph Jogging: 2.6 ± 0.5 (MET) Self-selected speed jogging: 3.6 ± 0.8 (MET) 3.0–3.5 mph medium-speed run: 5.2 ± 0.6 (MET) Optional medium-speed run: 4.6 ± 0.6 (MET) Fast run: 7.2 ± 1.6 (MET) Optional fast-speed run: 6.5 ± 1.7 (MET)	Pearson correlation coefficient	NR r = 0.69 * Activity type: slow (approximately 2.0 mph), medium (3.0–3.5 mph), and fast (run or walk of one’s choice); self-paced indoor walking at three speeds (slow, medium, and fast), up and down two flights of stairs, and a carrying task in which participants were asked to walk 10 m, pick up a 5-pound box, walk 10 m to place the box on a table, and walk back to the initial starting point
Kemp et al., 2020 [31]	NR Rest: 0.0 ± 0.0 (cpm) Sitting: 0.0 ± 4.1 (cpm) Stand: 0.0 ± 1.0 (cpm) Jogging: 2131.0 ± 1413.0 (cpm) Jogging: 7601.8 ± 3497.5 (cpm) Medium-speed running: 7509.1 ± 2106.6 (cpm) Fast running: 74.8 ± 685.8 (cpm) Fast kinetic cycling: 891.5 ± 1509.1 (cpm) Medium-speed walking: 3206.1 ± 720.1 (cpm)	Rest: 1.1 ± 0.1 (MET) Sitting: 1.2 ± 0.1 (MET) Stand: 1.2 ± 0.2 (MET) Jogging: 3.3 ± 0.6 (MET) Jogging: 6.8 ± 1.4 (MET) Medium-speed running: 8.3 ± 1.9 (MET) Fast running: 4.9 ± 1.8 (MET) Fast kinetic cycling: 6.8 ± 3.2 (MET) Medium-speed walking: 6.0 ± 1.2 (MET)	Spearman’s correlation coefficient	Rest: ρ = 0.14 Sitting: ρ = 0.32 * Standing: ρ = 0.17 Jogging: ρ = 0.59 * Jogging: ρ = 0.57 * Medium-speed running: ρ = 0.44 * Fast running: ρ = −0.08 Dynamic cycling: ρ = −0.04 Medium-speed walking: ρ = 0.36

Note: NR = not reported; VA = vertical axis; VM = vector magnitude; mph = mile/hour; cpm = counts/minute; cps = counts/second; MET = metabolic equivalent; * *p* < 0.01, there was a significant correlation.

**Table 5 sensors-23-08545-t005:** Descriptive analysis.

Authors and Year	Indices	Outcomes	Effect Size	Equation
Morris et al., 2019 [38]	Pearson’s correlation coefficient; mean deviation (90% CI)	All activities r = 0.74 *; MD = −37.6 (90% CI: −44.2, −30.9)	All activities Hip ES = 10.02	NR
Calabró et al., 2014 [32]	Pearson’s correlation coefficient (95% CI)	All activities r = 0.80 * (95% CI:0.65, 0.89)	All activities ES = −7.39	Freedson Combination (1998)
Mcminn et al., 2013 [41]	Mean Error (95% CI)	Jogging Hip ME = 0.77 (95% CI: 0.04, 1.51) Wrist ME = 1.22 (95%CI: 0.45, 1.99) Fast running Hip ME = −1.90 (95%CI: −2.77, 1.04) Wrist ME = −0.96 (95%CI: −1.77, 0.14)	Jogging Hip ES = 0.76 Wrist ES = 1.58 Fast running Hip ES = −1.59 Wrist ES = −0.95 All activities Hip ES = 0.18 & Wrist ES = −0.21	Freedson VM3 (2011)
Anastasopoulou et al., 2014 [34]	Pearson’s correlation coefficient; intra-group correlation coefficient; percentage error	Walking r = 0.53 *; ICC = 0.23; PE = 36.0% Fast walking r = 0.70 *; ICC = 0.35. PE = 31.3% Jogging r = 0.78 *; ICC = 0.70 # PE = −10.5% Walking uphill/downhill r = 0.84 *; ICC = 0.52 PE = 26.4% Downstairs r = 0.70 *; ICC = 0.24 PE = −37.5% Upstairs r = 0.41; ICC = 0.14; PE = 53.9%	Walking ES = 1.33 Fast walking ES = 1.24 Jogging ES = −0.49 Walking uphill/downhill ES = 1.01 Downstairs ES = −2.01 Upstairs ES = 1.49 All activities Hip ES = 0.22	Freedson VM3 (2011)
Kossi et al., 2021 [42]	Intra-group correlation coefficient	Riding Ankle ICC = 0.40 Hip ICC = 0.42 Wrist ICC = 0.42 Walking Ankle ICC = 0.31 Hip ICC = 0.69 # Wrist ICC = 0.32 Running Ankle ICC = 0.22 Hip ICC = 0.59 Wrist ICC = 0.21	Riding Ankle ES = 2.14 Hip ES = −2.33 Wrist ES = −2.79 Walking Ankle ES = 0.87 Hip ES = −1.29 Wrist ES = −1.48 Running Ankle ES = −2.03 Hip ES = −1.27 Wrist ES = −3.52 All activities Ankle ES = 0.33 Hip ES = − 0.92 Wrist ES = −1.72	Freedson Adult (1998)
Chang et al., 2019 [43]	Intra-group correlation coefficient; mean percentage error	0% angle treadmill running ICC = 0.877 #; MPE = 2.27 3% angle treadmill running ICC = 0.755 #; MPE = 10.85 6% angle treadmill running ICC = 0.504; MPE = 20.97	0% 5.61 km/h ES = 0.34 7.20 km/h ES = −0.23 8.02 km/h ES = −0.17 & 3% 5.61 km/h run ES = −1 7.20 km/h ES = −1.28 8.02 km/h ES = −1.15 6% 5.61 km/h ES = −1.98 7.20 km/h ES = −2.36 8.02 km/h ES = −2.11 All activities Hip 0% ES = −0.03 & Hip 3% ES = −0.67 Hip 6% ES = −1.04	Freedson VM3 Combination (2011)
Florez-Pregonero et al., 2017 [35]	Mean percentage error	Standing, reading MPE = 32.48% Sitting, typing MPE = 12.96% Sitting, playing chess MPE = 21.67% All sedentary activities MPE = 22.22% 1.62 km/h running MPE = −29.88% 2.41 km/h running MPE = −27.42% 3.24 km/h running MPE = −15.71% Cleaning the kitchen MPE = −11.20% All light activities MPE = −21.15%	Standing, reading ES = 2.24 Sitting, typing ES = 1.46 Sitting, playing chess ES = 2.19 All sedentary activities ES = 1.99 1.62 km/h running ES = 2.95 2.41 km/h running ES = 2.64 3.24 km/h running ES = 1.26 Cleaning the kitchen ES = 0.85 All light activities ES = 1.08 All activities Hip ES = −0.29	Freedson Adult (1998)
Schneller et al., 2015 [33]	Root Mean Square Error	Lying RMSE = 0.11 Sitting RMSE = 0.04 Standing RMSE = 0.10 4 km/h walking RMSE = 0.32 5 km/h walking RMSE = 0.57 6 km/h walking RMSE = 1.20 7 km/h running RMSE = 2.33 8.2 km/h running RMSE = 2.62 9.5 km/h running RMSE = 3.29 50 steps/min up the stairs RMSE = 4.06 70 steps/min up stairs RMSE = 4.74 90 steps/min on the stairs RMSE = 6.20 Riding1 RMSE = 21.89 Riding2 RMSE = 30.52 Riding3 RMSE = 40.50	Lying ES = 1.91 Sitting ES = 2.26 Standing ES = 3.86 4 km/h walking ES = 1.83 5 km/h walking ES = 2.43 6 km/h walking ES = 2.37 7 km/h running ES = 0.22 8.2 km/h running ES = 0.06 & 9.5 km/h running ES = 0.17 & 50 steps/min upstairs ES = 5.85 70 steps/min upstairs ES = 7.05 90 steps/min upstairs ES = 6.24 Riding1 ES = −14.31 Riding2 ES = −11.67 Riding3 ES = −9.05 All activities Hip ES = −0.54	Crouter Adult (2012)
Gastin et al., 2018 [37]	Mean error (95% CI); percentage error; root mean square error	4 km/h running ME = 27.4 (95% CI: −33.0, 87.8); PE = 25.3%; RMSE = 40.8 8 km/h running ME = 38.0 (95%CI: −20.7, 96.8); PE = 16.83%; RMSE = 48.1 12 km/h running ME = −41.2(95%CI: −90.2, 7.9); PE = −14.03%; RMSE = 47.9 Cycle training 1 ME = 127.2 (95% CI: −208.7, 45.8); PE = −56.93%; RMSE = 133.6 Cycle training 2 ME = −137.7 (95% CI: −214.6, 60.8); PE = −61.33%; RMSE = 143.0 Cycle training 3 ME = −132.7 (95% CI: −231.6, 33.8); PE = −59.33%; RMSE = 141.6	Hip 4 km/h running ES = 0.87 8 km/h running ES = 0.91 12 km/h run ES = 0.87 Cycle training 1 ES = 2.67 Cycle training 2 ES = 3.01 Cycle training 3 ES = 2.78	Freedson VM3 Combination (2011)
Thomson et al., 2021 [36]	Mean error (95% CI); mean percentage error; mean absolute percentage error	5.4 km/h running ME = 0.5 (95% CI: −2.1, 3.2); MPE = 6.5%; MAPE = 23.7% 6.5 km/h running ME = 0.2 (95%CI: −2.5, 2.8); MPE = 2.3%; MAPE = 14.0% 8 km/h running ME = 0.9 (95% CI: −2.9, 4.7); MPE = 11.1%; MAPE = 18.1%	5.4 km/h running ES = 1.40 6.5 km/h running ES = 0.04 8 km/h running ES = 0.44 All activities Hip ES = 0.16 &	Freedson VM3 (2011)

Note: The validity of the Actigraph was assessed using different statistical measures. The simple correlation coefficient was used to determine the validity in some studies (*), while the intra-group correlation coefficient was used in others (#). Additionally, the effect size was employed to assess the validity of the Actigraph in certain studies (&).

**Table 6 sensors-23-08545-t006:** Calculation equations [39,45,47,48,49,50,51,52,53,54,55].

Model	Equation	Model	Equation
Freedson Combination [48]	If cpm > 1951 kcals/min = 0.00094 × cpm + (0.1346 × weight − 7.37418) else kcals/min = cpm × 0.0000191 × weight	Puyau [52]	AEE (kcal·kg^−1^·min^−1^) = 0.0183 + 0.000010 × cpm
Freedson VM3 [49]	If VMcpm > 2453 kcals/min = 0.001064 × VM + 0.087512 × weight − 5.500229	Freedson [53]	METs = 2.757 + (0.0015 × cpm) − (0.000038 × cpm × age)
Freedson Adult [48]	MET = 1.439008 + (0.000795 × cpm)	Schmitz [54]	EE (kj·min^−1^) = 7.6628 + 0.1462 ([cpm − 3000]/100) + 0.2371 × weight − 0.00216 ([cpm − 3000]/1002) + 0.004077 ([cpm − 3000]/100 × weight)
Freedson VM3 Combination [49]	If VMcpm > 2453 kcals/min = 0.001064 × VM + 0.087512 × weight − 5.500229 else kcals/min = cpm × 0.0000191 × weight	Mattocks [55]	EE (kj·min^−1^·min^−1^) = −0.933 + 0.000098 × cpm + 0.091 × age − 0.04 × sex
Sasaki [49]	METs = 0.000863 × VM + 0.668876	Crouter REG-VA [45]	If VA/5s ≤ 35, EE = 1 child-MET If VA/5s > 35, EE (child-MET) = 1.592 + (0.0039 × VA/5s)
Crouter Vector Magnitude 2-Regression [39]	If VM/10s < 75, then EE = 1.0 METRMR. If VM/10s > 75 and VM/10s’CV < 25, then EE (METRMR) = 0.0137 (exp (0.848 × (ln (VM/10s)))) or VM/10s’CV > 25, then EE (METRMR) = 1.219 − (0.145 × (VM/10s))) − (0.0586(ln (VM/10s))) + (0.0229(ln (VM/10s)))	Crouter REG-VM [45]	If VM/5s ≤ 100, EE = 1.0 child-MET If VM/5s > 100, EE (child-MET) = 1.475 + (0.0025 × VM/5s)
Crouter Vertical Axis 2-Regression [39]	If VA/10s < 25, then EE = 1.0 METRMR. If VA/10s > 25 and VA/10s’CV < 35, then EE (METRMR) = 1.982 (exp (0.00101 × VA/10s)) or VA/10s’CV > 35, then EE (METRMR) = 2.842 + (0.00288 × VA/10s)	Work-Energy *	kcals = cpm × 0.0000191 × weight
Freedson Child *	EE = (0.0191 × cpm) − (0.000671 × cpm^2^) + (0.128 × weight) + (6.78 × sex) − 7.28	Santos-Lozano VT [47]	METs = 3.14153 + 0.00057 × VAcpm –0.01380 × weight − 0.00606 × age
Treuth [50]	METs = 2.01 + 0.000856 × cpm	Santos-Lozano VM [47]	METs = 2.7406 + 0.00056 × VMcpm–0.008542 × age − 0.01380 × weight
Trost [51]	EE (kcal/min) = −22.23 + 0.0008 × cpm + 0.08 × weight		

Note: cpm = counts/minute; VM = vector magnitude; VA = vertical axes; MET = metabolic equivalent; RMR = resting metabolic rate; sex = (male = 0, female = 1); CV = coefficient of variation. * Although the reference paper is the source of this formula, the formula itself was never published in a paper. The formula can be found at https://actigraphcorp.my.site.com/support/s/article/What-is-the-difference-among-the-MET-Algorithms (accessed on 19 September 2023).

**Table 7 sensors-23-08545-t007:** Accuracy of different calculation equations.

Authors and Year	Indices	Outcomes	Activities
Crouter et al., 2013 [39]	Effect size	All activities ES = 0.22 Crouter Vector Magnitude 2-Regression Model (2012) ES = 0.25 Crouter Vertical Axis 2-Regression Model (2012) ES = 0.35 Freedson Child Equation (2005) ES = 0.38 Treuth Equation (2004) ES = 0.27 Trost Equation (1998) ES = 0.16 Puyau Equation (2002) *	Lying down to rest, playing on computer, playing board games, cleaning, boxing, wall ball, walking, running
Zhu et al., 2013 [44]	Effect size	All activities ES = 0.30 Trost Equation (1998) ES = −0.31 Freedson Equation (2005) ES = 0.76 Puyau Equation (2002) ES = −0.30 Treuth Equation (2004) ES = 0.23 Schmitz Equation (2005) ES = −0.47 Mattocks Equation (2007)	3–8 km/h walk and run, dance and youth morning exercise, treadmill activities
Crouter et al., 2015 [45]	Effect size	All activities ES = 0.05 Crouter REG-VA (2015) * ES = 0.11 Crouter REG-VM (2015) *	Sitting still, watching TV, surfing the Internet, reading, board games, walking slowly, sitting at home, exercising, etc.
Kim et al., 2016 [46]	Effect size	All activities ES = 2.72 Crouter Vector Magnitude 2-Regression Model (2012) ES = 3.54 Crouter Vertical Axis 2-Regression Model (2012) ES = 2.76 Freedson Equation (2005) ES = 2.47 Trost Equation (1998) ES = 6.34 Puyau Equation (2002) ES = 3.21 Treuth Equation (2004)	12 randomly selected from 24 activities to complete
Santos-Lozano et al., 2013 [47]	Bias	All activities Bias = 0.07 Work-Energy (1998) Bias = −0.17 Freedson Combination (1998)	Rest, 3–9 km/h speed walking or running
Aguilar-Farias et al., 2019 [40]	Mean percentage absolute deviation	All activities MAPE = 8.6% Freedson VM3 (2011) MAPE = 38.6% Santos-Lozano VT (2013) MAPE = 32.8% Santos-Lozano VM (2013) MAPE = 13.8% Sasaki (2011)	Sitting, lying, walking, cleaning, doing housework

Note: * energy expenditure equations are considered valid.

## Data Availability

The data in the study are available in the included literature.

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
