# Peer review of "Validity of Actigraph for Measuring Energy Expenditure in Healthy Adults: A Systematic Review and Meta-Analysis"

_sensors, 2023, doi:10.3390/s23208545_

Round 1

Reviewer 1 Report

In the provided manuscript, the authors present a systematic review and meta-analysis aimed at assessing the validity of Actigraph triaxial accelerometry sensors in measuring energy consumption among healthy adults. The study's structure and findings are commendable, but a few minor revisions are recommended to enhance clarity and precision.

Abstract:

In the abstract, consider providing a brief summary of the methods employed and the key findings. This will assist readers in quickly grasping the scope and outcomes of the study.

Introduction:

In the introduction, the authors aptly outline the controversy surrounding the validity of Actigraph devices in measuring energy consumption. To further contextualize the issue, it would be beneficial to include a succinct overview of the practical implications of accurately measuring energy consumption and its relevance to health and fitness assessments.

Methods:

In the "Methods" section, specify the search criteria and keywords used for retrieving relevant literature from Pubmed, Web of Science, and Sportdiscuss databases. This will aid in reproducibility and transparency of the study.

Provide a brief explanation of the Downs and Black checklist and COSMIN checklist, highlighting the key aspects assessed by each checklist. This will elucidate the readers about the quality assessment process.

Results:

In the "Results" section, consider providing a concise summary of the characteristics of the included studies, such as study design, sample size, and participant demographics. This will offer readers a quick overview of the studies under consideration.

When reporting the meta-analysis results, clarify that the standardized mean difference (SMD) was calculated, along with its interpretation as a measure of effect size. Also, rephrase the presentation of the confidence interval (CI) to clearly convey its range.

Conclusion:

Revisit the conclusion to ensure that the conveyed message aligns with the study findings. As per the meta-analysis, the results do not show a significant difference between Actigraph and indirect calorimetry method. Therefore, the wording should reflect this absence of significant difference, rather than referring to a "significant correlation." Also, emphasize that the activity count can serve as a predictor for energy consumption, based on the meta-analysis outcomes.

Language and Grammar:

Perform a thorough proofreading to rectify minor grammatical and syntax errors throughout the manuscript. This will enhance the overall readability and professionalism of the paper.

Addressing these suggested revisions will likely enhance the clarity, accuracy, and overall quality of the manuscript. Great job on the study and best of luck with the revision process!

Perform a thorough proofreading to rectify minor grammatical and syntax errors throughout the manuscript. This will enhance the overall readability and professionalism of the paper.

Author Response

1.“In the abstract, consider providing a brief summary of the methods employed and the key findings. This will assist readers in quickly grasping the scope and outcomes of the study.”

Response: Thank you for your feedback. In the abstract section of the text, we have provided additional information regarding the methodology employed and the primary findings obtained. For the revised section, please consult lines 17-28 and 35-40.

2.”In the introduction, the authors aptly outline the controversy surrounding the validity of Actigraph devices in measuring energy expenditure. To further contextualize the issue, it would be beneficial to include a succinct overview of the practical implications of accurately measuring energy expenditure and its relevance to health and fitness assessments.”

Response: We appreciate your kind advice. In the introduction section, we have underscored the significance of precise measurement of physical activity energy expenditure in the realm of health, highlighting its practical value. For further information, please consult lines 44-59 and 87-115.

3.“In the "Methods" section, specify the search criteria and keywords used for retrieving relevant literature from Pubmed, Web of Science, and Sportdiscuss databases. This will aid in reproducibility and transparency of the study.Provide a brief explanation of the Downs and Black checklist and COSMIN checklist, highlighting the key aspects assessed by each checklist. This will elucidate the readers about the quality assessment process.”

Response: Thank you for your advice. The search strategies and results for each database have been systematically recorded and organized in a tabulated format. Please consult Table 1, specifically lines 126-129. For the Down and Black and COSMIN models, we have included supplementary explanations in the form of partial explanations. These additional explanations can be found in reference lines 150-170.

4.“In the "Results" section, consider providing a concise summary of the characteristics of the included studies, such as study design, sample size, and participant demographics. This will offer readers a quick overview of the studies under consideration.When reporting the meta-analysis results, clarify that the standardized mean difference (SMD) was calculated, along with its interpretation as a measure of effect size. Also, rephrase the presentation of the confidence interval (CI) to clearly convey its range.”

Response: We appreciate your kind advice. Regarding the overall attributes of the research, certain content has been omitted in order to enhance conciseness. Please make reference to lines 235-243.In the preceding text, we have provided an explanation regarding the concept of Standardized Mean Difference (SMD) as the outcome measure derived from a meta-analysis. Furthermore, we have also rephrased the confidence interval associated with SMD,refer to 281-293.

5.Revisit the conclusion to ensure that the conveyed message aligns with the study findings. As per the meta-analysis, the results do not show a significant difference between Actigraph and indirect calorimetry method. Therefore, the wording should reflect this absence of significant difference, rather than referring to a "significant correlation." Also, emphasize that the activity count can serve as a predictor for energy expenditure, based on the meta-analysis outcomes.

Response: Thank you for your advice. The conclusion of the paper has been revised. Please refer to line 514-530.

Reviewer 2 Report

This research was carried out by the guidelines outlined in the Preferred Reporting Items for Systematic Reviews and Meta-Analyses (PRISMA). The studies that were encompassed needed to fulfill these specific conditions: (1) conducting experiments where both Indirect calorimetry and Actigraph triaxial accelerometers were used simultaneously, (2) examining the variations, associations, and concurrences between the Actigraph device and Indirect calorimetry, (3) involving participants who were in a state of good health, capable of independent movement, and without any impediments to physical activity, (4) being published in the English language. Studies involving populations with health issues were deliberately left out. This work is exciting and can be helpful for many applications. After reading the manuscript, here are my comments:

  1. Please clarify the novelty and contribution of this work. How does the approach proposed by the authors differ from similar approaches in researching validity testing of sensors in measuring activity-related energy consumption in healthy individuals? There are many papers published that study these issues. What is your difference from the previous works and innovation? Clearly state the exact novelty of this study at the end of the introduction section.
  2. Motivation for the study must be given in the introduction section. What is the knowledge gap bridged by this study? More is needed for motivation, including references. Please clarify this issue.
  3. The way this work is formatted and presented is subpar. Kindly enhance it.
  4. Please revise Figure 1.
  5. The reference format needs to be revised; for example, ".....Calabró et al. (2014) [34] found that although energy consumption calculated.....".

Moderate editing of the English language is required. 

Author Response

1.“Please clarify the novelty and contribution of this work. How does the approach proposed by the authors differ from similar approaches in researching validity testing of sensors in measuring activity-related energy consumption in healthy individuals? There are many papers published that study these issues. What is your difference from the previous works and innovation? Clearly state the exact novelty of this study at the end of the introduction section.”

Response: Thanks for your advice. The novelty of this study lies in its integration of multiple studies comparing IC and Actigraph, which allows for more reliable conclusions to be drawn. By employing IC as the criterion, the research findings become more persuasive. Refer to line 92-115.

2.”Motivation for the study must be given in the introduction section.  What is the knowledge gap bridged by this study?  More is needed for motivation, including references.  Please clarify this issue.” 44-15

Response: We appreciate your kind advice. At lines 44-59, the motivation for the study is elucidated.  The research contributions are elucidated in the range of lines 92-115.

3.“The way this work is formatted and presented is subpar. Kindly enhance it.”

Response: Thank you for your advice.The format in which work is presented has been revised wherever possible.

4.“Please revise Figure 1.”

Response: We appreciate your kind advice. Figure 1 has been revised.

  1. The reference format needs to be revised; for example, ".....Calabró et al. (2014) [34] found that although energy consumption calculated.....".

Response: Thank you for your advice.The reference format has been revised.

Reviewer 3 Report

First of all, I appreciate the opportunity of reviewing this manuscript. Despite its overall merit, I have some concerns and comments that I copied below. I hope these topics help to improve the manuscript. 

Abstract

Remove the numbers between parentesis. 

Add the period or data that literature search was carried out. If possible, I reinforce the need for providing, at least, the terms used in the search strategy. 

Add the total amount of citations found and also the duplicates removal. 

Regarding conclusion, I suggest that the authors be more concise, followed by the main pratical implications of these findings. 

Overall, the authors must ensure proper formatting. For instance, the citations must be check, as well as English writting. 

Introduction

The first paragragh can be merge with the second. The authors can be more concisive when presenting the relevance of this topic for literature. Thus, I recommend revising these two paragraphs carefully. 

I suggest bring the disadvantages for using calorimetry in a separate paragraph. Based on them, The authors could introduce alternative tools, indicating how to cope with these disadvantages. Another point is advocate for advantages and disadvantages of Actigraph, which can enhance the relevance of this manuscript. 

From my perspective, the manuscript as a review lacks applicability statements when considering the scope of this study. In addition, it is essential for the reader to understand for whom, when, how use these tools (calorimetry or accelerometry).

The literature gap should be mention, followed by the purpose of this review, as the last paragraph. 

Methods

Did the authors previous register the review? 

Unfortenately, I diseagree with the topics presented in this section. When taken into account the study design, I suggest that the authors strictly follow the PRISMA. For instance, the Subsection could be entitle "Source information and search strategy". Therefore, I am looking for a version aligned with proposed in PRISMA. Additionally, I miss the search strategy applied for each database. I also did not find supplementary material. Am I correct? Or is somenthing missing?

Results

I have no comments about this section, except for formatting that should be carefully check. 

I also recommend the authors provide more adequate titles and footnotes for each table and figure. It is also important to double-check the order of data presentation. For instance, table 6 shoud be place previous in the results following the chronological steps conducted in this review. 

Discussion

I have no comments for this section, except for adding a topic named Practical implications. I recommend the authors to emphazise the implications for research. 

Conclusion

I suggest the authors to provide a more straight-forward conclusion. Any other results rather than the main finding must be properly address throughout the text instead of appearing once again in this section.  

I strongly recommend a double check on English writing, as well as formatting.

Author Response

1.“Remove the numbers between parentesis. Add the period or data that literature search was carried out. If possible, I reinforce the need for providing, at least, the terms used in the search strategy. Add the total amount of citations found and also the duplicates removal. Regarding conclusion, I suggest that the authors be more concise, followed by the main pratical implications of these findings. Overall, the authors must ensure proper formatting. For instance, the citations must be check, as well as English writting. ”

Response: We appreciate your kind advice. The search terms and strategies have been organized and presented in Table 1.We have made revisions to the conclusion, as well as the writing and reference format. Please consult lines 35-44 for reference.

2.“The first paragragh can be merge with the second. The authors can be more concisive when presenting the relevance of this topic for literature. Thus, I recommend revising these two paragraphs carefully. I suggest bring the disadvantages for using calorimetry in a separate paragraph. Based on them, The authors could introduce alternative tools, indicating how to cope with these disadvantages. Another point is advocate for advantages and disadvantages of Actigraph, which can enhance the relevance of this manuscript. From my perspective, the manuscript as a review lacks applicability statements when considering the scope of this study. In addition, it is essential for the reader to understand for whom, when, how use these tools (calorimetry or accelerometry).The literature gap should be mention, followed by the purpose of this review, as the last paragraph. ”

Response: Thank you for your advice. The article has been revised. Refer to line 48-119.

3.“Did the authors previous register the review? Unfortenately, I diseagree with the topics presented in this section. When taken into account the study design, I suggest that the authors strictly follow the PRISMA. For instance, the Subsection could be entitle "Source information and search strategy". Therefore, I am looking for a version aligned with proposed in PRISMA. Additionally, I miss the search strategy applied for each database. I also did not find supplementary material. Am I correct? Or is somenthing missing?”

Response:Thank you for your advice. Unfortunately, this review is not registered.  We have compiled the search strategy and results for each database in Table 1. And we have added the PRISMA Guide as Addtional file 1. The operating principles of PRISMA are mainly to guide the writing process of the article, and there should be no restrictions on how to write the title.

4.“I have no comments about this section, except for formatting that should be carefully check. I also recommend the authors provide more adequate titles and footnotes for each table and figure. It is also important to double-check the order of data presentation. For instance, table 6 shoud be place previous in the results following the chronological steps conducted in this review. ”

Response:I really appreciate your advice. We have made modifications to the placement of Tables 5 and 6. Are there any additional specific requirements for the footnote?

  1. “I have no comments for this section, except for adding a topic named Practical implications. I recommend the authors to emphazise the implications for research. ”

Response: Thank you for your advice. The article has been revised. Please refer to line 482-505.

  1. “I suggest the authors to provide a more straight-forward conclusion. Any other results rather than the main finding must be properly address throughout the text instead of appearing once again in this section.  ”

Response: I really appreciate your advice. The article has been revised. Please refer to line 537-552.

Reviewer 4 Report

Summary

Overall, the paper is well-written and the method of creating the review is clear.

Minor points:

1.      Instead of stating “one article…, two articles…”, it is better to add references and makes it easier to find which articles were discussed.

2.      There are a lot of similarities between Fig 2 and 3. Could you combine the two?

3.      The format and style of reference throughout the paper is different. Please check and use the same format of the references in the manuscript.

4.      It would be helpful to add a paragraph of what is the future trend of using Actigraph in such applications, and what type of research is not well conducted or needs more exploration.  

5.      There are so many consumer products and research prototypes that have the same functionality. Could you add a paragraph to compare Actigraph to different ones and talk about the benefits of using Actigraph?

English is good; minor editting is needed. 

Author Response

1.“ Instead of stating “one article…, two articles…”, it is better to add references and makes it easier to find which articles were discussed.”

Response: We appreciate your kind advice. We have included the references at the munuscript. Please review them. The writing method of one and two articles is mainly to reflect the quantitative characteristics.

  1. There are a lot of similarities between Fig 2 and 3. Could you combine the two?

Response: Thanks for the advice. Figure 2 has many similarities to Figure 3, but these are two different models. One is a random model and the other is a fixed model. I think showing the two figures under two separate subheadings can present the results better.

  1. The format and style of reference throughout the paper is different. Please check and use the same format of the references in the manuscript.

Response: Thank you for your advice. The format of the references has been revised.

  1. It would be helpful to add a paragraph of what is the future trend of using Actigraph in such applications, and what type of research is not well conducted or needs more exploration.

Response: Thank you for your advice. We have conducted a examination of the future research trends, which can be found in the reference line 499-505.

  1. There are so many consumer products and research prototypes that have the same functionality. Could you add a paragraph to compare Actigraph to different ones and talk about the benefits of using Actigraph?

Response: Very grateful for your advice. Unfortunately, this is a shortcoming of this study. Because the Actigraph is currently the more popular device for measuring physical activity, this study only addressed the validity of the Actigraph. Future studies should also be conducted to evaluate the measurement validity of similar devices, as well as studies that compare between multiple devices.

Round 2

Reviewer 2 Report

The authors have replied to a significant part of my previous comments. The paper is almost ready for publication, but there are some aspects that need to be improved. Already informed you, but it hasn't been revised yet.

1. The reference format needs to be revised; for example, ".....Calabró et al. (2014) [33] found that although energy consumption calculated.....".

Just minor editing of the English language is required.

Author Response

The reference format needs to be revised; for example, ".....Calabró et al. (2014) [33] found that although energy consumption calculated.....".

Response: We appreciate your kind advice. The manuscript has been revised